# BLOB-Based AOMs: A Method for the Extraction of Crop Data from Aerial Images of Cotton

**Andrew Young** [1,2]**, James Mahan** [2,]*****, **William Dodge** [1] **and Paxton Payton** [2]

[1] Department of Plant and Soil Science, Texas Tech University, 2500 Broadway, Lubbock, TX 79409, USA, andrew.young@ttu.edu (A.Y.), william.dodge@ttu.edu (W.D.)

[2] Cropping Systems Research Laboratory, Agricultural Research Service, United States Department of Agriculture, 3810 4th Street, Lubbock, TX 79415, USA, paxton.payton@usda.gov

***** Correspondence: james.mahan@ars.usda.gov; Tel.: +1-806-723-5221

**Abstract:** The use of aerial imagery in agriculture is increasing. Improvements in unmanned aerial systems (UASs) and the hardware and software used to analyze imagery are presenting new options for agricultural studies. One of the challenges associated with improving crop performance under water deficit conditions is the increased variability in the growth and development inherent in low water settings. The nature of plant growth and development under water deficits makes it difficult to monitor the response to environmental changes. Small field and plot-level experiments are often variable enough that averages of seasonal crop characteristics may be of limited value to the researcher. This variability leads to a desire to resolve fields on finer temporal and spatial scales. While UAS imagery provides an ability to monitor the crop on a useful temporal scale, the spatial scale is still difficult to resolve. In this study, an automated computer software framework was developed to facilitate resolving field and plot-level crop imagery to finer spatial resolutions. The method uses a Binary Large Object (BLOB)-based algorithm to automate the generation of areas of measurement (AOMs) as a tool for crop analysis. The use of the BLOB-based system is demonstrated in the analysis of plots of cotton grown in Lubbock, Texas, during the summer of 2018. The method allowed the creation and analysis of 1133 AOMs from the plots and the extraction of agronomic data that described plant growth and development.

**Keywords:** UAS; Aerial Imagery; BLOB; cotton; water deficit; variability

---

## 1. Introduction

Agricultural fields are an orderly collection of plants that promote efficient and equal access to resources. Efficient crop management is based upon meeting the resource demands of a crop in the proper amounts at the proper times.

Variation from the desired populations and distribution of plants within a field can reduce yield as compared to a more uniform planting [1–4]. Understanding the resource demands of the crop (radiation, water, and nutrition), and planting in a manner that provides for those needs, is an essential part of modern agriculture.

While a uniform crop stand is a goal at planting, variation in emergence and seedling survival is common and the result is the non-uniformity of crop stands. Replanting to achieve uniformity is an option but producers are often uncertain about the value of replanting, since, if the replant does not increase yield, it is seen as waste of time and resources. Data-based replant decisions require information on the extent of stand variability and the potential effect of that variability on yield, perhaps one hundred days later.

Cotton is a crop whose perennial nature results in a flexible growth habit, with a relatively flexible relationship between in-field plant density and yield [1]. This flexibility complicates stand assessments and replant decisions [3–7].

The optimal seeding rates and the crop outcomes for cotton have been the subject of numerous studies [4–10]. In spite of those efforts to understand the relationship between stand variability and yield, the question of what constitutes an "acceptable stand" in rainfed cotton remains complex and difficult to determine under field conditions. New tools and approaches might be useful for the required detection and characterization of crop growth [11] and development in variable stands.

Aerial images of crops provide a useful perspective regarding identifying and quantifying crop uniformity [6,11,12]. Recent improvements in tools for aerial imaging in an agricultural setting have led to the increased use of Unmanned Aerial Systems (UASs) in research and production environments [13–21]. UAS measurements make it possible to view not only fields and plots, but individual plants [13,14] and individual components of single plants [15–17]. This improved measurement efficiency presents challenges in terms of the amount of data that can be collected and perhaps, most importantly, how these data streams can be analyzed in an agriculturally relevant setting [18–21].

Image analysis in crops generally involves the extraction of plant information that is associated with field locations represented in images.

In an aerial image of a crop, plants comprise a very small fraction of the image at emergence and, by the end of the season, comprise a larger fraction of the image data. To focus an analysis on the plant-containing pixels, the plant pixels must be separated from the background. An area of measurement (AOM) is a defined collection of pixels that delineates a region of interest in the image. AOMs can be defined in a variety of image analysis programs (e.g., ImageJ, GIMP). As the number and size of images and the number of AOMs desired increases, the manual creation of AOMs can become time consuming and a source of error itself. In variable cotton stands the number of AOMs required to describe the variability can become large and computer automation of the AOM creation process becomes increasingly more valuable. A Binary Large Object (BLOB) is a collection of binary data stored as a single entity [22]. BLOBs, which can be generated using a computer algorithm, can delineate a collection of contiguous pixels and can be used for the creation of AOMs.

In order to reduce the effects of environmental stresses on cotton, it is important to understand the patterns, severity, and mechanisms of stress on seasonal timeframes. The collection of field data is an obstacle to the development of an actionable understanding of yield losses due to stress. The ability to automate crop monitoring over time will allow the collection and analysis of field data on a greater scale.

The goal of this study was the development of a BLOB-based protocol to create and define AOMs for the extraction of agronomic data from cotton aerial images. Open source software was used in the system to enhance user access and reduce cost.

## 2. Materials and Methods

### 2.1. Agronomic Information

Upland Cotton (*Gossipium hirsutum L.*) was grown in small plots in the summer of 2018 in Lubbock, Texas (33°34′40″ N, 101°53′24″ W). All plots were planted on a 1 m row spacing around a center pivot. Planting dates were March 15 for plot 1, April 12 for plot 2 and May 29 for plot 3. Plots were rainfed with 248 mm in plot 1, 214 mm in plot 2 and 364 mm in plot 3. Conventional tillage was used, while herbicide and insecticide application followed established best practices.

2.1.1. Plot Information, Seed, and Plant Counts

Plot area (m²) was defined as the planted area of the field. Plots were hand measured before planting and marked with flags in the field. After planting, UAS flights were flown and plots were measured digitally using geographic information system (GIS) software. All plots were seeded at a rate of 100,000 seeds/ha. UAS data was collected on June 5 and 14, with a 75% image overlap and 75%

image sidelap at a height of 20 m. The lighting conditions were variable, with minimal shadowing behind the plants in both flights. These flights were composited into orthomaps, which were later used for digitized hand counting in GIS software. Digitized counts were made by placing a point layer in the GIS software and later totaling the points placed in that layer. This was done for all plots in the study. Total planted areas of the plots were used for calculations of seeds/plot based on the 100,000 seeds/ha seeding rate.

### 2.1.2. Maximum Canopy Calculations

UAS data was collected on August 13, with a 75% image overlap and 75% image sidelap at a height of 50 m for plot 1. The lighting was optimal, with minimal shadowing and near solar noon. UAS data was collected September 6 and 11, with a 75% image overlap and 75% image sidelap at a height of 35 m for plots 2 and 3. The lighting on September 6 was optimal ambient light conditions, with no shadowing on the crop. The September 11 flight was taken early in the morning, with large shadowing effects on the crop. The cotton crop, at this point in the season, had reached physiological maturity with initial fruiting sites entering the boll opening stage. The UAS data collected on these flights was later used for cotton plant canopy isolation and further image analysis. Final canopy area, as percentage of plot size, was calculated using the extracted plant canopy from these orthomaps. The extracted plant canopy area was calculated using built-in area calculations of polygons in the GIS software. The area calculated was then divided by the total planted area for the plot to determine the final canopy area (% of plot size).

### 2.1.3. Maturity and Yield

The maturity date of each plot, defined as 60% open bolls, was 140 days after planting (DAP) for plot 1, 129 DAP for plot 2, and 143 DAP for plot 3. Maturity dates were determined through visual inspection of all plots. All yield is reported as seed cotton weight. The plot-level yields were obtained using a 4 row cotton stripper in December. Yield in BLOBs was estimated from an aerial imaging algorithm. The yield algorithm was developed on white pixel counts in a total of eighty 1-meter spaces that were subsequently hand harvested. UAS data was collected on October 4 and 26, with a 75% image overlap and 75% image sidelap at a height of 35 m and 30 m respectively. The lighting conditions of the orthomaps were variable, with a more ambient occluded light on October 4 and a near solar noon flight on October 26. These flights were used for aerial yield estimation in plot 1. UAS data was collected on November 9, with a 75% image overlap and 75% image sidelap at a height of 35 m. This flight was used for aerial yield estimation in plots 2 and 3. The flight was near solar noon with some light shadowing on the crop.

### 2.2. UAS Imaging System and Image Processing

The image capture platform used was a DJI Phantom 4 Pro UAS. This UAS was equipped with the stock original equipment manufacturer (OEM) 20 megapixel RGB camera (DJI Industries, Shenzhen, China). At the 20 megapixel resolution, the ground sample distance (GSD) varied from a minimum of 0.59 cm/pixel at 20 m to a maximum of 2.95 cm/pixel at 100 m above ground level (AGL). The camera on this device was mounted to a gimbal that utilizes an automatic stabilization feature to ensure that, at the time of image capture, the camera is facing directly towards the ground. A down-facing camera at the time of image capture results in a nadir view in the raw image data which facilitates the orthomap creation process.

UAS image data was collected at the USDA-ARS Plant Stress and Water Conservation Laboratory in Lubbock, Texas (33° 35' 40.85"N, 101° 54' 4.99"W). The cotton crop in the scene was part of a rain-grown cotton matrix, experiment where entries have both a unique planting date and water treatment [23]. The matrix experiment is comprised of entries planted over a range of planting dates and the aerial imagery used in this study was collected according to the protocol of the matrix experiment.

2.2.1. UAS Image Processing

UAS data processing was conducted using an in-house, open-source software framework that allows for end-to-end UAS data collection and analysis. The workflow for processing aerial image data begins with the act of collecting raw image data with an UAS. All flights were planned with the DroneDeploy flight planning software (DroneDeploy mobile ver. 4.0.0). Once the flight plan has been generated, the flight mission is conducted resulting in an image data set typically comprised of several hundred images. These raw image data sets from each flight mission were then composited into singular, high-resolution, 2D orthomaps with OpenDroneMap [24]. OpenDroneMap (ODM) is an open-source mapping software that allows users to produce both 2D and 3D maps of a given scene. The resultant orthomaps are imported into QGIS (QGIS ver. 3.8.1) [25] which is an open-source geographic information system (GIS) software. The accuracy and alignment from one orthomap to the next was evaluated and corrected through the native georeferencing capabilities in QGIS. The observed level of precision with respect to the consistent alignment of objects in successive orthomap scenes is ±5 cm. The AOMs were hand digitized in QGIS to identify the individual plantings in the orthomap scenes. This was performed by creating a vector layer in QGIS that served as a polygon mask. That polygon mask allows for the raster data that falls within the polygon space to be extracted. In other words, the orthomap raster layers were "clipped" using polygons and individual plantings were extracted for further analyses.

*2.3. BLOB-Based AOM Creation*

The planting-level extracted AOMs served as the basis for the binary large object (BLOB) analysis. BLOB detection is a technique that utilizes one of many algorithms to facilitate the detection and extraction of connected pixels in binary image data. Python (Python ver. 3.7) [26] and the OpenCV (OpenCV ver. 4.0.0) [27] library provide several ways to isolate connected pixels in such data. The BLOB-based AOMs are formed after connected pixel areas are extracted from the binary image and serve as a basis for plant-level extractions of data on single or groups of plants. BLOB-based AOM detection in this research was implemented through a three-part process: (1) cotton plant canopy isolation, (2) image thresholding, and (3) BLOB identification/extraction.

2.3.1. Cotton Plant Canopy Isolation

The growth of cotton in this experiment was evaluated and characterized by observing the nature of canopy development as captured in a series of orthomap raster layers managed in QGIS. The maximum canopy size and the extent to which said canopy is congruous over the length of a single row are two primary cotton growth characteristics that defined the analysis in this research. Because these two characteristics are aspects of the cotton crop canopy, the first phase of the BLOB analysis was the task of separating the cotton canopy from the rest of the scene in each of the three plantings. This task was performed by evaluating red, green, and blue (RGB) pixel intensities for all pixels in each of the three planting-level AOM extractions. Images were not adjusted for color calibration other than auto-white balancing from the UAS camera system. The logic driving the canopy extraction was to use Python and OpenCV to create a mask comprised solely of pixels where the green band (G) pixel intensity was greater than both the blue (G>B) and red band (G>R) pixel intensities (G>B, G>R). This is a simple yet effective approach to isolate the green pixels in the scene that represent living biomass such as the plant canopy. The mask is then applied to the planting-level AOM extraction, effectively removing all elements of the scene that are not plant related.

After the cotton canopy has been isolated, the planting-level AOM extractions were subjected to a thresholding process that was implemented with the binary thresholding methods native to the OpenCV library. Binary thresholding is a process that converts all pixel intensity values in a single channel image to either the minimum (0) or maximum (255) possible values. All extracted AOM image data in this research was eight-bit data, resulting in a bit-depth of 256 ($2^8$). The range of possible pixel intensities in this case is 0–255 inclusive, and so binary thresholding will result in a single channel image, where all pixel intensities are either 0 or 255. The thresholding process requires the

user to provide a thresholding value that acts to determine whether a pixel will either be thresholded to 0 or 255 depending on whether the pixel intensity is either above or below the threshold. This threshold was determined by visually comparing the plant to the background scene. If the pixel intensity is below the threshold value, then the binary thresholding process will set that pixel to 0, otherwise, in the case were the pixel is above the threshold, the value will be set to 255. This process results in a single-channel, black and white, binary image where all pixel intensities are either 0 (black) or 255 (white). The image threshold is a result of the plant canopy isolation and serves to isolate green canopy. Elements in the scene that are determined to be green are set to 255, and everything else below the previous canopy isolation threshold is set to 0.

### 2.3.2. BLOB Identification and Extraction

After the planting-level AOM extractions had been masked and thresholds applied, the final step was to programmatically isolate and extract the BLOB-based AOMs. BLOB identification/extraction was performed with the cv2.findContours method native to OpenCV. This algorithm [28] identifies a curve that joins all continuous points along a contoured boundary and retrieves them from the image. With planting-level AOM extractions masked and subjected to binary thresholding, the boundary between the white BLOBs and the black background is pronounced and easily detectable. As a result, the cv2.findContours method returns a list of all contour data. The contoured data was used to iteratively extract data from the plantings for each BLOB-based AOM. The contours and respective label data were saved to a file. Along with individual BLOB-based AOMs, the entire set of BLOB-based AOMs for each planting were collectively extracted and saved. These BLOB-based AOM sets, for each planting, were imported back into QGIS and georeferenced to ensure alignment with the existing orthomap data managed in QGIS. This alignment results in a QGIS vector layer of polygons representing all BLOB-based AOMs in each of the three plantings. Figure 1 shows the workflow of the UAS image processing and yield estimation process using BLOB-based AOMs.

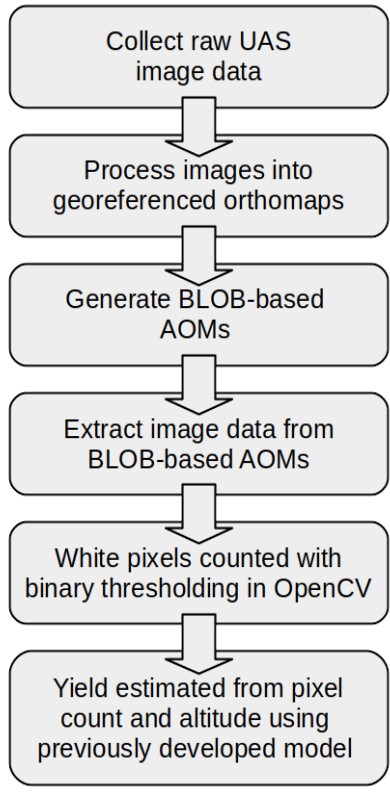

**Figure 1.** Flowchart of the unmanned aerial systems (UAS) image processing and yield estimation with a binary large object based area of measurement (BLOB-based AOM) data.

## 3. Results and Discussion

### 3.1. Agronomic Plot-Level Data

The planting density was approximately 10 seeds/meter of row (100,000 seeds/ha) and at a 100% emergence value the total number of plants/plots could be 7450 for plot 1, 8656 for plot 2 and 9920 for plot 3. Plot 1 was 992 m² and had 9920 seeds planted, plot 2 was 866 m² and had 8656 seeds planted, and plot 3 was 745 m² and had 7450 seeds planted based on plot area. Digitized hand counts of plants showed that plot 1 had 1107 plants, plot 2 had 1598 plants, and plot 3 had 3067 plants. Table 1 shows the number of plants in each planting based on digitized hand counting at 60, 54, 45 DAP for plots 1, 2, and 3 respectively. The number of planted seeds that germinated and emerged by 30 DAP varied with planting date with the highest % in the latest planting (41% in plot 3).

Days required to reach maximum canopy was relatively consistent at approximately 110 DAP. Maximum canopy area varied from 189 to 268 m². Maturity dates among the plot varied by 13 days.

The plot-level yield in the three plots was obtained using a 4 row cotton stripper in December. Total seed cotton yield was 1353 kg/ha in plot 1, 1179 kg/ha in plot 2, and 944 kg/ha in plot 3.

Table 1. Plot-level agronomic data for three plantings.

| Plot | Plant Date | Plot Area (m²) | Seeds | Plants | Plants (% of seed) | Days to Maximum Canopy | Maximum Canopy Area (m²) | Final Canopy Area (% of plot Size) | Days to Maturity | Yield Seed Cotton (kg/ha) |
|---|---|---|---|---|---|---|---|---|---|---|
| 1 | March 15 | 992 | 9920 | 1107 | 11% | 112 | 193 | 19.4 | 140 | 1353 |
| 2 | April 12 | 866 | 8656 | 1598 | 18% | 108 | 268 | 30.96 | 129 | 1179 |
| 3 | May 29 | 745 | 7450 | 3067 | 41% | 110 | 189 | 25.43 | 143 | 944 |

### 3.2. Plot-Level Imagery Analysis

Images were filtered to extract green pixels that represent plant material. Figure 2 shows pre-processed orthomap plot-level AOM images of the three plots taken near the end of the season. Stand uniformity in the plots was highly variable at the end of the season. Much of the variation is a result of early versus later planting dates with plots 1 and 2 planted in late spring (March 15 and April 12) and plot 3 was planted on May 29. The images in Figure 2 were acquired at 153 DAP for plot 1, 147 DAP for plot 2, and 100 DAP for plot 3. The total planted row area (one-meter row spacing) for each plot was 992 m² for plot 1, 866 m² for plot 2, and 745 m² for plot 3.

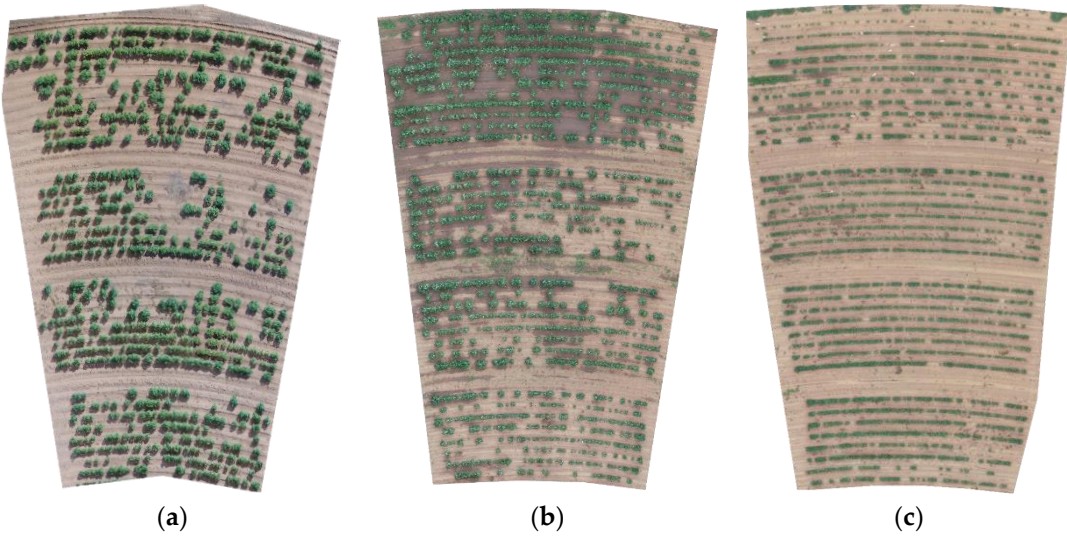

| (a) | (b) | (c) |

**Figure 2.** Red, green, and blue (RGB) orthomap image of three cotton plots. Images were acquired near the end of the growing season and show the maximum extent of canopy area. Panel (**a**), plot 1, was planted on March 15, 2018 and imaged 153 days after planting (DAP). Panel (**b**), plot 2, was planted on April 12, 2018 and imaged 147 DAP. Panel (**c**), plot 3, was planted on May 29, 2018 and imaged 100 DAP.3.3. Maximum Canopy Size and Analysis.

Final canopy area (% of plot size) was calculated using the extracted plant canopy at initial stages crop maturity (Table 1). Estimated final total canopy cover for each plot is 193 m² for plot 1 at 112 DAP, 268 m² for plot 2 at 108 DAP and 189 m² for plot 3 at 110 DAP. As a fraction of the total planted area in each plot, the final canopy cover was 19.4% in plot 1, 30.96% in plot 2 and 25.43% in plot 3.

### 3.4. Summary of Plot-Level Image Analysis

The analysis of the plot-level images (Table 1) provided the basis for assessments of crop characteristics including; 1) final canopy area (m², measured ground cover fraction at maturity), 2) days to maximum canopy area (m², ground cover fraction over time series), 3) plant population (from manual counts), and yield (estimated). Within-plot information may provide additional information on plant performance needed to understand the crop responses to factors such as soil variability, seedling vigor, weed pressure and germplasm choices. Finer spatial resolution, such as individual plant level resolution, is needed to address these issues.

### 3.5. Automated BLOB-Based AOM Creation

A BLOB-based approach was developed to define AOMs for data extraction and analysis in each of the three plots. These geo-registered BLOB-based AOMs provide the ability to extract plant data from spatiotemporal imagery collected from planting to harvest.

Virtually all in-season crop information occurred within the areas of the field delineated by the end-of-season canopy AOMs. The end-of-season AOMs can be applied backward in time on image sequences from the start to the end of the growing season.

In terms of the extraction of data from a sequence of images, the accuracy of the data returned for each pixel in an image or in an AOM is dependent on the precision attained in the geo-registration process. In this analysis, the location of a given AOM in a series of images was subject to positional variation of ±5 cm. The 5 cm positional resolution of an AOM should be viewed in light of the fact that plant canopies are biological and not rigidly defined objects. Canopies are commonly subject to random movement in the range of 5 cm due to wind and other biotic factors. Additionally, for cotton, heliotropic and water status-induced leaf movements are common.

Image-to-image shifting of an AOM in an image can result in "clipping" of the AOM near an edge that can result in exclusion of the plant material from the analysis. In terms of the reliable extraction of data from AOMs applied to seasonal image sequences, the inclusion of bare soil beyond the outside of the AOM (plant material) does not result in errors in the measurement of characteristics of plants within the AOM. This potential source of variability can largely be accounted for by oversizing the AOM by an amount that will offset "movement" of the AOM in sequential images.

The end-of-season plot images with the resulting BLOB-based AOMs are shown in Figure 3. A total of 1133 BLOBs were defined in the three plots. The number of BLOBs varied among the plots, with a total of 317 BLOBs in plot 1, 391 BLOBs in plot 2 and 425 BLOBs in plot 3. Table 2 shows summary characteristics of the BLOBs for each plot.

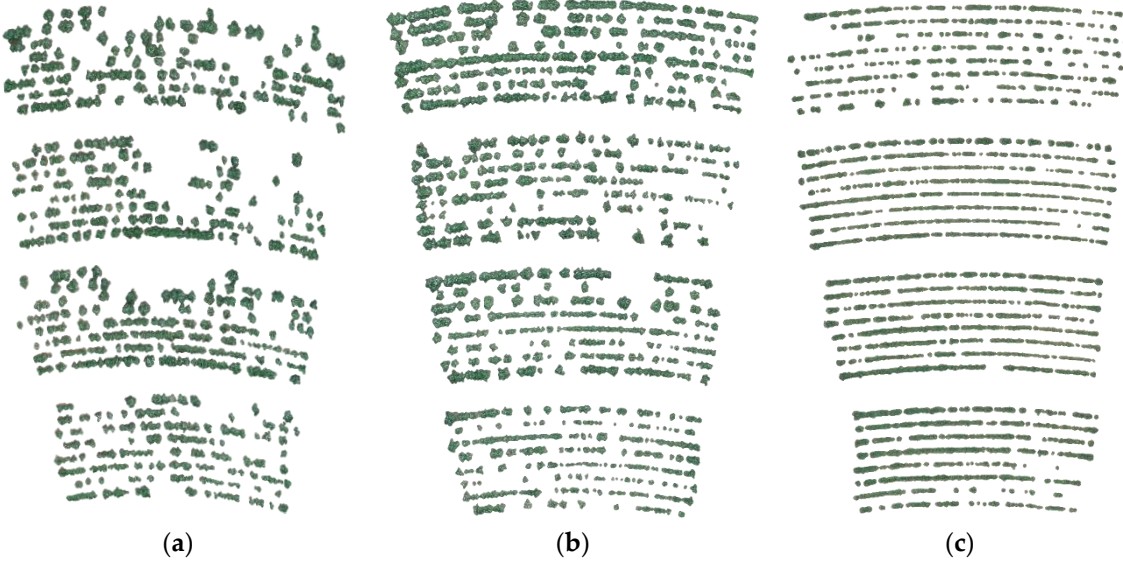

(**a**)　　　　　　　　　　(**b**)　　　　　　　　　　(**c**)

**Figure 3.** Results of the application of the binary large object (BLOB) algorithm to the orthomap red, green, and blue (RGB) images of the three plots. A total of 1133 BLOBs were defined in the three plots. The number of BLOBs varied among the plots, with a total of (**a**) 317 BLOBs in plot 1, (**b**) 391 BLOBs in plot 2 and (**c**) 425 BLOBs in plot 3.

**Table 2.** Summary statistics of binary large objects (BLOBs) for three plantings.

| Plot Number | Number of BLOBs | Mean BLOB Size (m²) | Minimum BLOB Size (m²) | Maximum BLOB Size (m²) |
|---|---|---|---|---|
| 1 | 317 | 0.93 | 0.12 | 6.95 |
| 2 | 391 | 0.69 | 0.03 | 9.55 |
| 3 | 425 | 0.45 | 0.04 | 4.640 |

*3.6. BLOB Color Mapping for Visualization*

Given the relatively large number of BLOB-based AOMs that can be extracted from field plots, the interpretation of the data can become potentially complex. The ability to visualize the plant characteristics as an initial step in interpretation is potentially useful. The production of color-coded images of AOMs in a GIS environment allows for the visualization of agricultural characteristics of the plants within BLOB-based AOMs.

Figure 4 shows the BLOB images for all three plots with yield indicated in color. The scale is common for all three plots and differences (and similarities) in yields between plots are evident. The generally higher yield in plot 1 versus plot 2 and the ranges of yield values are evident. Figure 5 shows a representation of within plot targeted analysis using BLOB-based AOMs in red (a) and a visual representation of the selected BLOB-based AOMs in panel (b). Figure 6 shows three plant characteristics of the BLOB with the values mapped as colors. Panel (a) shows yield (g) per BLOB, panel (b) shows yield (g) per plant and panel (c) shows the number of plants per BLOB. This approach makes it possible to visualize the variation in the plants within a BLOB with relative ease. While the three images in Figure 6 represent a single point in in the season, it is possible to visualize the changes in BLOBs over time by applying color mapping to sequential images.

The data in Figures 4–6 demonstrate the extraction of yield and plant number data from BLOB-based AOMs. Using this method, the rates of change in crop characteristics such as flowering, canopy cover, and height over a season and could be visually represented as well.

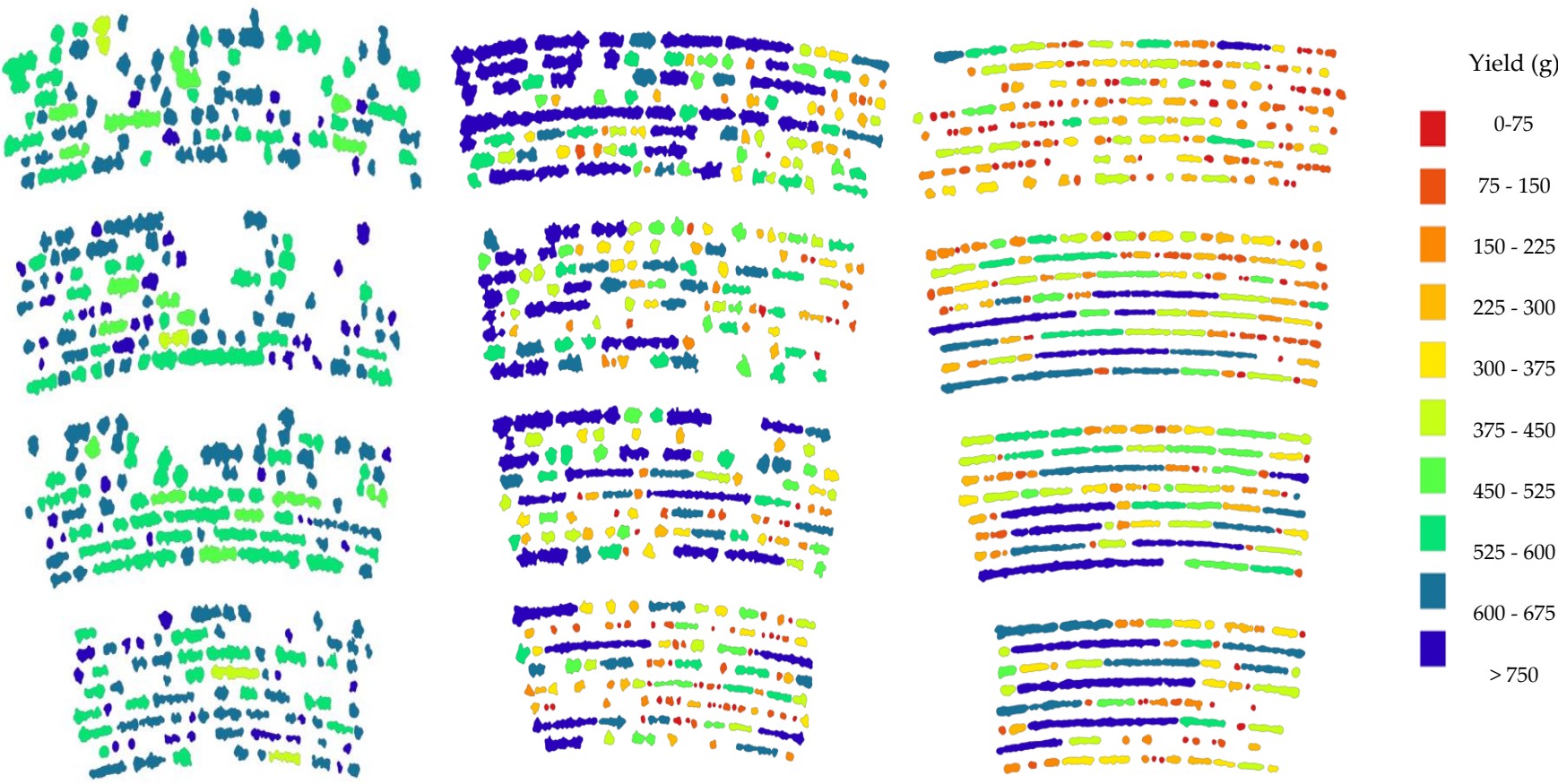

**Figure 4.** Binary large object (BLOB) image of plots 1, 2, and 3 with the yield (g/BLOB) mapped as BLOB color. The same symbology is assigned to all three plots.

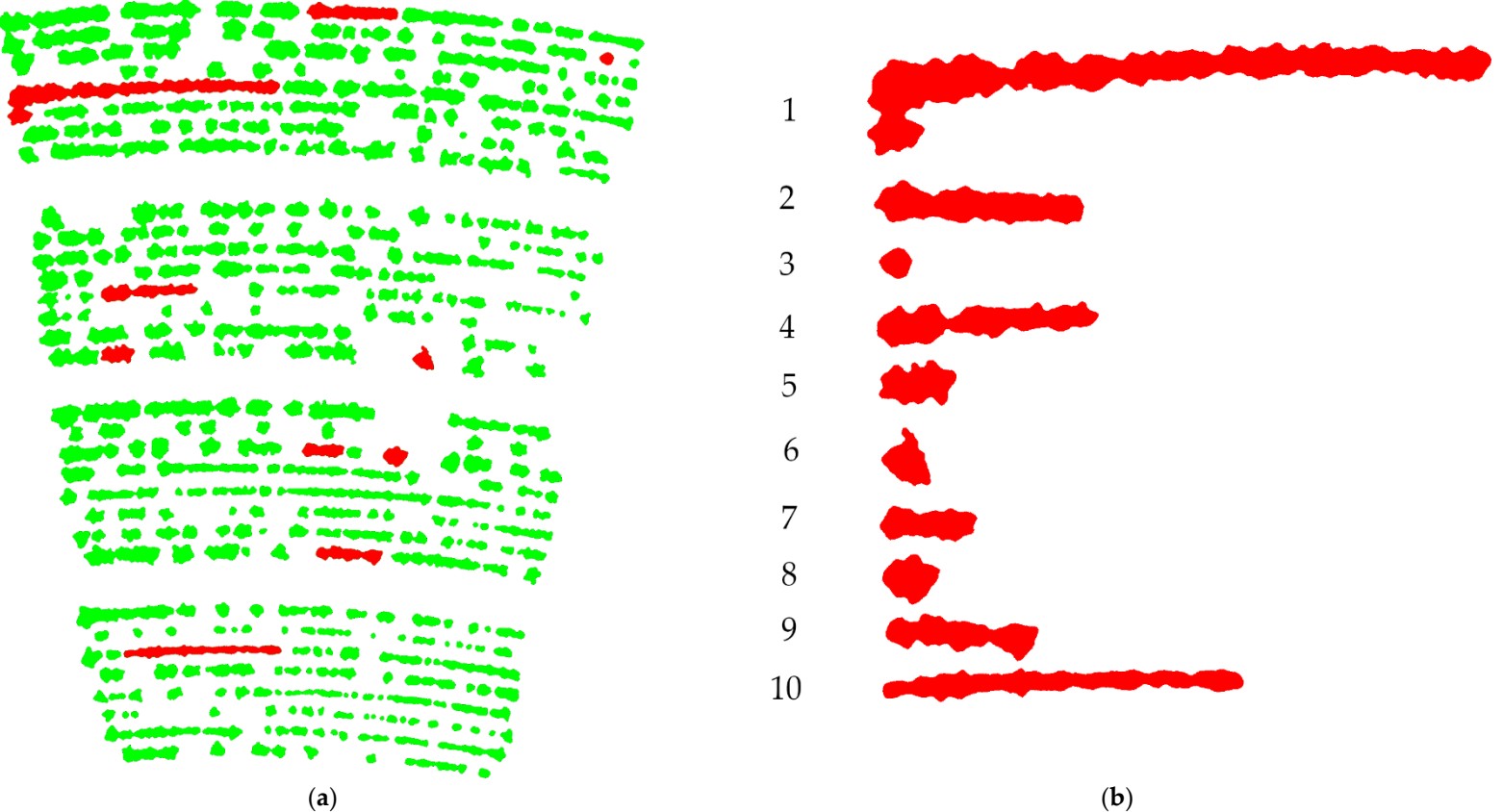

**Figure 5.** Panel (**a**), subset of binary large object-based areas of measurement (BLOB-based AOMs) selected for data extraction from plot 2. BLOBs were chosen to represent a variety in terms of size and distribution within the plot. Panel (**b**), shows the BLOBs for comparative purposes.

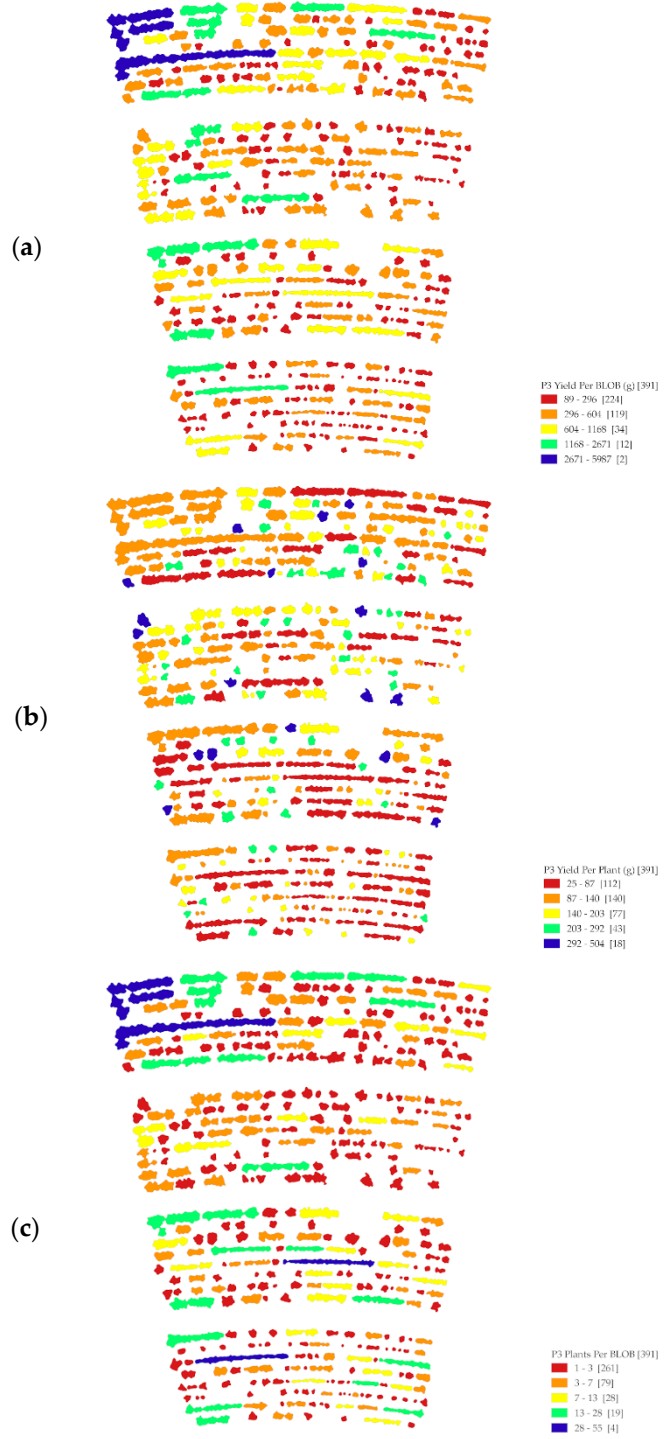

**Figure 6.** Binary large objects (BLOBs) mapped with color to indicate extracted plant characteristics. (**a**) Yield (g) per BLOB; (**b**) yield (g) per plant; (**c**) number of plants per BLOB.

*3.7. BLOB Data Extraction*

The ability to resolve a crop into groupings of plants in a field at a resolution of 5 cm² is potentially useful as a tool for monitoring the growth and development down to the single plants and groups of plants on a field scale. The distribution of end-of-season BLOB-based AOMs provides the ability to go back in time by monitoring the development of a plant, or group of plants, across a series of sequential images.

To demonstrate the extraction of crop data from sequential BLOB-based AOMs, 10 BLOBs in plot 2 were chosen for analysis. Figure 5 shows total of 10 BLOBs that were chosen to represent a variable range of the BLOBs in plot 2 with respect to size and shape. The 10 BLOBs chosen for analysis are highlighted in red. The goal was to create a collection of BLOBs that were representative of the variation in the image. Some were isolated round BLOBs of relatively small size (BLOB #3 with one plant in 0.33 m²). Others were oblong of varying lengths and sizes (BLOB #1 with 55 plants in 9.55 m²). The following characteristics, shown in Table 3, were associated/extracted from the 10 BLOBs; area (m²), plants/BLOB, time from planting to maximum canopy area, yield (kg/ha), and yield per plant (g/plant).

**Table 3.** Targeted analysis of BLOBs in planting 2.

| BLOB | Area (m²/BLOB) | Plants/BLOB | Plants/m² | Days to Maximum Canopy | Canopy Expansion (cm²/day/plant) | Yield (g/BLOB) | Yield/m² (g) | Yield (g/plant) |
|---|---|---|---|---|---|---|---|---|
| 1 | 9.55 | 55 | 5.7 | 108 | 16 | 3353 | 351 | 60 |
| 2 | 2.74 | 19 | 6.9 | 108 | 13 | 1613 | 587 | 85 |
| 3 | 0.33 | 1 | 3.0 | 102 | 32 | 182 | 552 | 182 |
| 4 | 2.61 | 11 | 4.2 | 109 | 22 | 1208 | 463 | 110 |
| 5 | 1.13 | 3 | 2.6 | 117 | 32 | 689 | 610 | 230 |
| 6 | 0.74 | 1 | 1.4 | 131 | 56 | 340 | 459 | 340 |
| 7 | 1.14 | 6 | 5.3 | 109 | 17 | 670 | 588 | 112 |
| 8 | 0.85 | 3 | 3.5 | 108 | 26 | 374 | 440 | 125 |
| 9 | 1.77 | 11 | 6.2 | 109 | 15 | 757 | 428 | 69 |
| 10 | 3.02 | 42 | 13.9 | 109 | 7 | 1457 | 482 | 35 |

The results in Table 3 demonstrate the extraction of crop characteristics from BLOB-based AOMs. The area (m²) of the BLOBs varied from 0.33 to 9.55. The number of plants per BLOB varied from 1 to 55 with the area of the BLOBs increasing with the number of plants per BLOB (Figure 7A) The number of days required for the canopy to reach its maximum area in each BLOB was generally similar and varied from 108 to 131 days at an average of 111 days (Figure 7B). The seasonal rate of canopy expansion varied among BLOBs and was a function of the number of plants in the BLOB (Figure 7C). As the number of plants in a BLOB increased the rate of canopy expansion declined. The yield as grams (g) for each BLOB varied from 182 g for BLOB 3 to 3353g for BLOB 1. The yield (g) as a function of the area of the BLOB (m2) (Figure 7D) was variable, with a mean of 523 g/m2, a minimum of 427 g/m2 in BLOB 9 and a maximum of 626 g/m2 in BLOB 1. The yield per plant (g) as a function of the number of plants in a BLOB varied across the BLOBs, with a mean of 140 g/plant, a minimum of 35 g/plant in BLOB 10, and a maximum of 340 g/plant in BLOB 4 (Figure 7E). These results demonstrate the data that can be extracted from the BLOB-based AOMs using this approach. It is possible that significant insights into plant responses to growth conditions could be derived from a BLOB analysis.

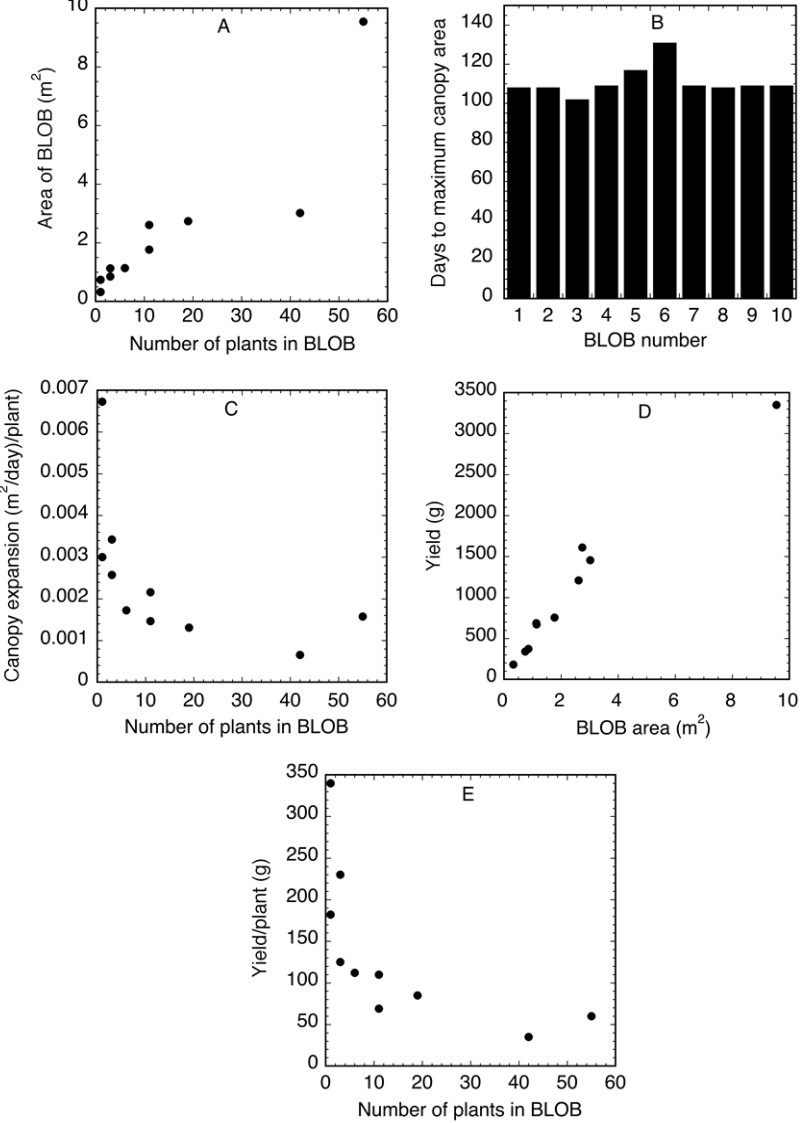

**Figure 7.** Agronomic data extracted from 10 binary large object-based areas of measurement (BLOB-based AOMs). (**A**) Area of a BLOB as a function of the number of plants per BLOB. (**B**) Number of days required for the canopy to reach its maximum area. (**C**) The seasonal rate of canopy expansion. (**D**) The yield (g) as a function of the area of a BLOB. (**E**) The per plant yield as a function of the number of plants in each BLOB.

## 4. Conclusions

In this study, the use of a computer framework to define relatively large numbers of AOMs, as BLOBs, based on aerial observations of the plants themselves has been demonstrated. The application of an automated BLOB-based AOM approach enhances the utility of a UAS, as a tool, to measure crop growth and development. The analysis of the plots using BLOB-based AOMs when compared to that based solely on AOMs drawn around the plot boundaries suggests that the increase in resolution could be useful, particularly in plots that are non-uniform.

This BLOB-based approach provides a means to resolve even relatively small plots of plants into hundreds of elements ranging in size from single to multiple plants. The creation of BLOBs in images taken at any point in time and applying those BLOBs to images from other times during a season allows for the efficient and reliable extraction of time-series data from a seasonal image sequence. The creation of BLOB-based AOMs using end-of-season images limits time series extractions to the areas of the field where there are plants at season's end and thus, where yield was produced. The use

of plant-based BLOBs allows the researcher to identify the relatively small number of objects of importance from the billions of observed pixels of data in an experimental plot.

Strengths of the approach include:

(1) BLOB generation can be automated to a great extent.
(2) The ability to define AOMs based on end-of-season canopy area provides a means to track growth and development from planting to harvest.
(3) The ability to assign and modify AOMs at any point in the season allows flexibility inherent to this method.

Weaknesses of the approach include:

(1) Automated AOM selection can produce many AOMs that may require complex data management approaches to utilize.
(2) The number of AOMs that can be extracted from uniform plots is reduced. In uniform plots, the AOMs may be limited to the number of rows.
(3) Between-row canopy closure results in AOMs that cross rows and reduces the scale of resolution.

This BLOB-based AOM approach may be a useful tool in agricultural image analysis. The ability to extract data at a single plant level over a season may provide valuable insight into crop responses to environmental variation in the field.

**Author Contributions:** This project was conceptualized by J.M., P.P. and A.Y.; the methodology was developed by J.M., A.Y., and W.D.; software code was created by A.Y. and W.D.; the original draft of the manuscript was prepared by J.M.; additional writing—review and editing was by, J.M., A.Y., P.P. and W.D. All authors have read and agreed to the published version of the manuscript.

**Funding:** This research was funded by USDA/ARS and Cotton Inc.

**Acknowledgments:** We acknowledge the assistance of Julia Brown in manuscript preparation and Avery Starkey in data collection.

**Conflicts of Interest:** The authors declare no conflict of interest. The funders had no role in the design of the study; in the collection, analyses, or interpretation of data; in the writing of the manuscript, or in the decision to publish the results. Mention of a trademark, warranty, proprietary product, or vendor does not constitute a guarantee by the USDA and does not imply approval or recommendation of the product to the exclusion of others that may be suitable. USDA is an equal opportunity provider and employer.

**Standard USDA disclaimer for discrimination:** The U.S. Department of Agriculture (USDA) prohibits discrimination in all its programs and activities on the basis of race, color, national origin, age disability, and where applicable, sex, marital status, familial status, parental status, religion, sexual orientation, genetic information, political beliefs, reprisal, or because all or part of an individual's income is derived from any public assistance program. USDA is an equal opportunity provider and employer.

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
