# Peer review of "BLOB-Based AOMs: A Method for the Extraction of Crop Data from Aerial Images of Cotton"

_agriculture, doi:10.3390/agriculture10010019_

Round 1

Reviewer 1 Report

Overall the paper has much improved quality and clarity. But there are still some minor concerns that have not been addressed. I would like to see an overall flowchart of the UAV image processing and yield estimation. Any color calibration performed onto acquired data? Sections 2.3.1 and 2.3.2 may be condensed into a single section on plant canopy segmentation. In Table 1, the column names "Maximum canopy (DAP)" and "Maturity (DAP)" are confusing. Do the terms refer to the DAP for Maximum canopy and maturity respectively?

Author Response

Reviewer 1

Overall the paper has much improved quality and clarity. But there are still some minor concerns that have not been addressed.

I would like to see an overall flowchart of the UAV image processing and yield estimation.

*Inserted UAV image processing and yield estimation flowchart. Line 211.

Any color calibration performed onto acquired data?

*Line 174 clarified no color calibrations

Sections 2.3.1 and 2.3.2 may be condensed into a single section on plant canopy segmentation.

*Condensed section into 1 section

In Table 1, the column names "Maximum canopy (DAP)" and "Maturity (DAP)" are confusing. Do the terms refer to the DAP for Maximum canopy and maturity respectively?

*Clarified in Table 1 titles

Reviewer 2 Report

I've read the manuscript. It has been improved a lot. The introduction and methodology are much clearer than the previous version. The only thing need to improve is the conclusion, which should provide brief take-away message. The conclusion can be shorten further by modifying the first several paragraphs (they are still background introduction), or move them into discussion.

Author Response

Reviewer 2

I've read the manuscript. It has been improved a lot. The introduction and methodology are much clearer than the previous version.

The only thing need to improve is the conclusion, which should provide brief take-away message.

The conclusion can be shorten further by modifying the first several paragraphs (they are still background introduction), or move them into discussion.

*Shortened the conclusion and inserted brief takeaway message at 403.

Round 2

Reviewer 2 Report

I suggest to accept the manuscript.

This manuscript is a resubmission of an earlier submission. The following is a list of the peer review reports and author responses from that submission.

Round 1

Reviewer 1 Report

Unmanned aerial systems (UAS) are gaining momentum for efficient acquisition of spatial and temporal data for precision management in crop production. This study reports on the use of UAS for monitoring growth of cottons, and the development of a software framework for extracting relevant agronomic data. This paper needs careful spell check to eliminate grammatical and wording errors. My specific comments for the paper are given as follows.

I suggest that for clarity and consistency the authors provide full names of the abbreviations (e.g., BLOB, AOM, UAS, DAP, …) when they first appear in the text. In the introduction Line 70-78, more strong justification is needed for the present study. You mention UAS is several times better than what was previously possible. But it remains unclear as of what was previous possible. UAS is efficient for large-scale mapping, but not necessary leads to finer scales of measurements. It is also not clear what level spatial resolutions are desired for the cotton research. The authors may point out the problems with previous methods (or practices) and then come to why and the objectives of this study. Lines 92-93 about acquisition of yield data, the yield data are used for later color mapping? Did you use the three methods together for obtaining ground truth? If so how could you do this with aerial imaging? What is the image size for the RGB camera? Please provide manufacture info for the camera? Line 130, change “Open Drone Map” into “OpenDroneMap”, and please provide citation information for the software (and manufacture info for QGIS). In L141-143 where BLOB is introduced, please provide a brief description of how the algorithm works, and please provide version info for OpenCV and Python that were used for BLOB detection. In Line 146, the cv2.findCountours() was used for BLOB detection? Regarding green pixel detections in Line 158-159, please gives explanations why this logic works for canopy extraction. Generally HSV space may be more preferred for color extraction or segmentation (based on hue values). Did you experiment with HSV space? Regarding thresholding in section 2.4, how was threshold determined? Trial and error tests or automatically chosen? I suggest the authors provide a flow chart of image process and analysis stems for acquired UAS image data. Double check two DAPs in Table 1. The images in Figure 1 are raw, unprocessed RGB images? If so, why are they nonrectangular? Any citations for Line 246 “AOMs are often created by hand ….”. The description of image error and spatial resolution in Line 267-268 is confusing. How was the error determined, at what level of height? Did you refer the error to as spatial resolution? The description “A resolution of canopy in the ….inevitable” also sounds problematic. Did you mean that is probably the finest achievable resolution? Double check 4th column data in Table 2. Since the paper is aimed at developing a computational framework, could you provide additional information about data processing and analysis efficiency of the framework?

Author Response

Reviewer #1

Unmanned aerial systems (UAS) are gaining momentum for efficient acquisition of spatial and temporal data for precision management in crop production. This study reports on the use of UAS for monitoring growth of cottons, and the development of a software framework for extracting relevant agronomic data. This paper needs careful spell check to eliminate grammatical and wording errors. My specific comments for the paper are given as follows.

I suggest that for clarity and consistency the authors provide full names of the abbreviations (e.g., BLOB, AOM, UAS, DAP, …) when they first appear in the text. Provided explanations of abbreviations before they appear in paper

In the introduction Line 70-78, more strong justification is needed for the present study. You mention UAS is several times better than what was previously possible. But it remains unclear as of what was previous possible. UAS is efficient for large-scale mapping, but not necessary leads to finer scales of measurements. It is also not clear what level spatial resolutions are desired for the cotton research. The authors may point out the problems with previous methods (or practices) and then come to why and the objectives of this study. Section re-written to clarify resolution

Lines 92-93 about acquisition of yield data, the yield data are used for later color mapping? Did you use the three methods together for obtaining ground truth? If so how could you do this with aerial imaging? Yield measurement section has been re-written for clarity

What is the image size for the RGB camera? Please provide manufacture info for the camera? Addressed in the UAS Image section

Line 130, change “Open Drone Map” into “OpenDroneMap”, and please provide citation information for the software (and manufacture info for QGIS). Fixed and provided citations

In L141-143 where BLOB is introduced, please provide a brief description of how the algorithm works, and please provide version info for OpenCV and Python that were used for BLOB detection.  Addressed in text

In Line 146, the cv2.findCountours() was used for BLOB detection?  Yes

Regarding green pixel detections in Line 158-159, please gives explanations why this logic works for canopy extraction. Generally HSV space may be more preferred for color extraction or segmentation (based on hue values).

Did you experiment with HSV space? Yes. Green pixel mask worked sufficiently for our study.

Regarding thresholding in section 2.4, how was threshold determined? Trial and error tests or automatically chosen? Simple binary threshold clarified in section 2.4.

I suggest the authors provide a flow chart of image process and analysis stems for acquired UAS image data.

Double check two DAPs in Table 1. Corrected in table

The images in Figure 1 are raw, unprocessed RGB images? No, these are plot level extractions from orthomosaic images.

If so, why are they nonrectangular? Plot are planted in circular orientation around a pivot irrigation system.

Any citations for Line 246 “AOMs are often created by hand ….”. Clarified in text

The description of image error and spatial resolution in Line 267-268 is confusing. How was the error determined, at what level of height? Did you refer to the error as spatial resolution? Clarified in text the spatial vs positional variation in plant canopy.

The description “A resolution of canopy in the ….inevitable” also sounds problematic. Did you mean that is probably the finest achievable resolution? Removed sentence in text

Double check 4th column data in Table 2. Cleaned up table data

Since the paper is aimed at developing a computational framework, could you provide additional information about data processing and analysis efficiency of the framework?  Refocused the paper on the BLOB-based approach rather than developing computational frameworks.

Reviewer 2 Report

General comments

The manuscript demonstrates an UAS image processing approach where individual groups of continuous vegetation (BLOBs) are identified and plant attributes are represented at that scale. Useful information about the approach is given and the process for BLOB extraction is well described. However, the manuscript has significant issues which prevent publication at its present form. At the moment, the manuscript is closer to a ‘scientific note’ or ‘short communication’ than to an original scientific study. The intro lacks a proper definition of the problem, a clear hypothesis and a proper characterization of the state of the art, along with (and most importantly) identifying the issues in that research domain. Much is said about the importance of accounting for spatial variability in the analysis of agricultural data. However, precision ag research has been dealing with this for many years already. One option (to frame the paper to the format of an original scientific study, not a communication note) would be for the authors to highlight issues with the current approaches being used to tackle that greater issue (the need to account for spatial variability in analysing agricultural data) using a comprehensive review and identifying a research gap which the proposed approach could potentially fill (a question worth asking is why would one use a BLOB approach over a map of pixels, for example, to represent the data at finer spatial resolutions?). Once the hypothesis is set, a methodology must be developed to test that hypothesis. Because that initial scientific framework was not given (the problem, justification and hypothesis), what follow the methods are loose arguments about the potential of the technique without a discussion over data that can actually demonstrate the benefit (or otherwise) of the proposed approach over others. Another significant issue is the lack of details about how field measurements were taken (the study is not replicable). More specific comments are given throughout the manuscript.

Overall, if the authors prefer to turn this into a short communication paper (my recommendation), the text needs to be significantly more cohesive and straightforward.

Author Response

Reviewer #2

The manuscript demonstrates an UAS image processing approach where individual groups of continuous vegetation (BLOBs) are identified and plant attributes are represented at that scale. Useful information about the approach is given and the process for BLOB extraction is well described.

However, the manuscript has significant issues which prevent publication at its present form. At the moment, the manuscript is closer to a ‘scientific note’ or ‘short communication’ than to an original scientific study.

The intro lacks a proper definition of the problem, a clear hypothesis and a proper characterization of the state of the art, along with (and most importantly) identifying the issues in that research domain.

Much is said about the importance of accounting for spatial variability in the analysis of agricultural data. However, precision ag research has been dealing with this for many years already. One option (to frame the paper to the format of an original scientific study, not a communication note) would be for the authors to highlight issues with the current approaches being used to tackle that greater issue (the need to account for spatial variability in analysing agricultural data) using a comprehensive review and identifying a research gap which the proposed approach could potentially fill (a question worth asking is why would one use a BLOB approach over a map of pixels, for example, to represent the data at finer spatial resolutions?).

Once the hypothesis is set, a methodology must be developed to test that hypothesis. Because that initial scientific framework was not given (the problem, justification and hypothesis), what follow the methods are loose arguments about the potential of the technique without a discussion over data that can actually demonstrate the benefit (or otherwise) of the proposed approach over others. Another significant issue is the lack of details about how field measurements were taken (the study is not replicable). More specific comments are given throughout the manuscript.

Overall, if the authors prefer to turn this into a short communication paper (my recommendation), the text needs to be significantly more cohesive and straightforward.

Refocused paper to represent this authors recommendations for a short communications paper.

Reviewer 3 Report

Summary
The study performed by Young et al. introduced a Binary Large OBject (BLOB) approach to improve the efficiency and accuracy of crop growth status monitoring by using Unmanned Aircraft System (UAS). With limited inputs, their approach is able to track crop growth from planting to harvest.

Board comments
1. The manuscript is not well structured. The object of this study is not clear. The introduction and methodology include a lot of information, but they are not organized logically, which is difficult for readers to focus on the major contribution of this work.

2. Although the authors explained a lot for their approach, it is hard for researchers to re-perform the study due to a lack of quantitative description.

3. There is no comparison with parallel works. Thus, I don't think it is proper to say that this approach is more 'robust'. Moreover, how can the authors prove that their method is more advanced without comparisons?

4. Based on my understanding, the method introduced in this study is not novel. The authors just used UAS to take a set of high-resolution photos. After preprocessing, a simple rule regarding the green band was applied to distinguish crop canopies. Then they painted those canopied by measured data (e.g. crop yield), which are shown in Figure 2-5. They are not novel.

Specific comments
-Major comments
1. The abstract should show the key findings of this research or the advantages of the new approach. However, too much research background is introduced in the abstract, which is not necessary. For example, to explain the complexity of plant growth, the authors mentioned "water deficit". However, it is not the major concern of this work. Moreover, the impacts of wet conditions on crops are not investigated in the result. So why mention this issue a lot?

2. Numerous abbreviations are used before an explanation. Please check "UAS", "BLOB", and "AOM" in the abstract, and other abbreviations in the main text.

3. Two paragraphs are used to introduce the early planting date and the water deficit. They explain why the authors set up 3 plots with different planting dates and rainfall amounts. However, there is rare analysis about the relations among planting date, wetness conditions, and crop status. I agree to increase crop variation for later comparison by varying planting dates and wet conditions, but it is not necessary to spend two paragraphs for detail discussion. So I recommend either reducing the description of planting date and water issues, or adding more results to investigate the effects of these two factors on crops.

4. Line 81-82: "The method is reliable, relatively fast and low-cost and offers increased flexibility when compared to other methods currently in use." It is not convincing without a literature review of other methods. Furthermore, in the discussion, the parallel comparison is missing either.

5. Line 95-111: This paragraph introduces the experiment design, not "UAS measurements". Please change the structure. Moreover, several long sentences exist in this paragraph, e.g, Line 100-102, 107-109. It is better to reform them to be simple but brief.

6. Sect 2.2-2.5: This part reads like an uncompleted technical manual, not a methodology of scientific papers. Firstly, the methodology should detailed describe the scientific method that can help readers to reproduce the study regardless of software. For instance line 183-184, what is the scientific approach behind the cv2.findContours()? Secondly, several key processes are explained by fuzzy sentences. For example line 156-157 and 160-161, how did you evaluate RGB pixel intensities? How to specify that "the green band pixel intensity is greater than the red and blue band"? Do you mean [green band intensity > red band intensity AND green band intensity > blue band intensity]? Or do you mean [green band intensity > sum of red and blue band intensity]?

7. Sect 3.2: How did you obtain the crop yield data? By measurement or by remote sensing-based estimation? How about other variables mentions in this section?

8. Line 238-239: How did you define the rate of canopy area development and plant population? Please provide the formulas and units.

9. The discussion is not complete. The strength and weakness of the approach should be further discussed in comparison to other similar works.

10. The conclusion is missing.

-Minor comments
1. Line 249: Is this the objective of this study? It may be better to move it in the introduction.

2. Line 260: "field delineated" -> "field is delineated"?

3. Table 2: Why show maximum BLOB size in column 4?

4. Line 298: "between" -> "among"?

5. Line 301: "Figure 3" -> "Figure 4"?

6. Line 302: "Figure 4" -> "Figure 5"?

7. Figure 4: Since Figure 4b is mentioned later than Figure 5, it is better to move it after the Figure 5.

8. Figure 6: Increase font size. Please use capital letters for BLOB to maintain consistency.

Author Response

Reviewer #3

Summary
The study performed by Young et al. introduced a Binary Large OBject (BLOB) approach to improve the efficiency and accuracy of crop growth status monitoring by using Unmanned Aircraft System (UAS). With limited inputs, their approach is able to track crop growth from planting to harvest.

Broad comments
1. The manuscript is not well structured. The object of this study is not clear. The introduction and methodology include a lot of information, but they are not organized logically, which is difficult for readers to focus on the major contribution of this work.

Rewrote a more focused introduction for clarity.

Although the authors explained a lot for their approach, it is hard for researchers to re-perform the study due to a lack of quantitative description.

Expanded on in materials and methods.

There is no comparison with parallel works. Thus, I don't think it is proper to say that this approach is more 'robust'. Moreover, how can the authors prove that their method is more advanced without comparisons?

Removed comparisons with other works to focus on methodology of paper.

Based on my understanding, the method introduced in this study is not novel. The authors just used UAS to take a set of high-resolution photos.

After preprocessing, a simple rule regarding the green band was applied to distinguish crop canopies. Then they painted those canopied by measured data (e.g. crop yield), which are shown in Figure 2-5. They are not novel.

Expanded our results and discussions with attention to how and why we use the BLOB-based approach.

Specific comments

-Major comments
1. The abstract should show the key findings of this research or the advantages of the new approach. However, too much research background is introduced in the abstract, which is not necessary. For example, to explain the complexity of plant growth, the authors mentioned "water deficit". However, it is not the major concern of this work. Moreover, the impacts of wet conditions on crops are not investigated in the result. So why mention this issue a lot?

Refocused this section on the method not the water impact and crop issues.

Numerous abbreviations are used before an explanation. Please check "UAS", "BLOB", and "AOM" in the abstract, and other abbreviations in the main text.

Fixed in text

Two paragraphs are used to introduce the early planting date and the water deficit. They explain why the authors set up 3 plots with different planting dates and rainfall amounts. However, there is rare analysis about the relations among planting date, wetness conditions, and crop status. I agree to increase crop variation for later comparison by varying planting dates and wet conditions, but it is not necessary to spend two paragraphs for detail discussion. So I recommend either reducing the description of planting date and water issues, or adding more results to investigate the effects of these two factors on crops.

Removed this section

Line 81-82: "The method is reliable, relatively fast and low-cost and offers increased flexibility when compared to other methods currently in use." It is not convincing without a literature review of other methods. Furthermore, in the discussion, the parallel comparison is missing either.

Moved to discussion

Line 95-111: This paragraph introduces the experiment design, not "UAS measurements". Please change the structure. Moreover, several long sentences exist in this paragraph, e.g, Line 100-102, 107-109. It is better to reform them to be simple but brief.

Rewrote this section for clarity.

Sect 2.2-2.5: This part reads like an uncompleted technical manual, not a methodology of scientific papers. Firstly, the methodology should detailed describe the scientific method that can help readers to reproduce the study regardless of software.

Rewrote this section for clarity.

For instance line 183-184, what is the scientific approach behind the cv2.findContours()? Secondly, several key processes are explained by fuzzy sentences. Clarified in section 2.3.3 and provided citation for cv2.findContours algorithm.

For example line 156-157 and 160-161, how did you evaluate RGB pixel intensities? How to specify that "the green band pixel intensity is greater than the red and blue band"? Do you mean [green band intensity > red band intensity AND green band intensity > blue band intensity]? Or do you mean [green band intensity > sum of red and blue band intensity]?

Clarified in section 2.3 with some pseudo-code showing green bands > red and blue bands.

Sect 3.2: How did you obtain the crop yield data? By measurement or by remote sensing-based estimation? How about other variables mentions in this section?

Clarified that the yield was obtained by machine harvest and that it was plot level yield. Section 2.1 clarified plot and BLOB yield methodology.

Line 238-239: How did you define the rate of canopy area development and plant population? Please provide the formulas and units.

See 3.5 Summary of plot-level image analysis

The discussion is not complete. The strength and weakness of the approach should be further discussed in comparison to other similar works.

Refocused the results section for results and discussion.

The conclusion is missing.

A conclusion was added

-Minor comments
1. Line 249: Is this the objective of this study? It may be better to move it in the introduction.

Line 260: "field delineated" -> "field is delineated"? Table 2: Why show maximum BLOB size in column 4? Line 298: "between" -> "among"? Line 301: "Figure 3" -> "Figure 4"? Line 302: "Figure 4" -> "Figure 5"?

Addressed all above comments and errors in text

Figure 4: Since Figure 4b is mentioned later than Figure 5, it is better to move it after the Figure 5.

Fixed Figure mentions

Figure 6: Increase font size. Please use capital letters for BLOB to maintain consistency.

Addressed this in text.